# The Role of microRNA Expression and DNA Methylation in HPV-Related Cervical Cancer: A Systematic Review

**DOI:** 10.3390/ijms252312714

**Published:** 2024-11-26

**Authors:** Alessandra Pulliero, Giulia Cassatella, Pietro Astuni, Zumama Khalid, Stefano Fiordoro, Alberto Izzotti

**Affiliations:** 1Department of Health Sciences, University of Genoa, 16132 Genoa, Italy; giuliacassatella@live.it (G.C.); zumama.khalid@gmail.com (Z.K.); stefano.fiordoro@edu.unige.it (S.F.); 2Department of Internal Medicine and Medical Specialties, University of Genoa, 16132 Genoa, Italy; pietro.astuni@gmail.com; 3Department of Experimental Medicine, University of Genoa, 16132 Genoa, Italy; izzotti@unige.it

**Keywords:** microRNA (miRNA), human papillomavirus (HPV), cervical cancer, methylation

## Abstract

Human papillomavirus (HPV) infection is a major etiologic factor in cervical cancer, a major cause of cancer-related morbidity and mortality among women worldwide. The role of microRNA (miRNA) dysregulation in cervical carcinogenesis is still largely unknown, but epigenetic changes, including DNA methylation and miRNA regulation, are crucial factors. The integration of HPV DNA into the host genome can lead to alterations in DNA methylation patterns and miRNA expression, contributing to the progression from normal epithelium to cervical intraepithelial neoplasia and, ultimately, to cervical cancer. This review aimed to examine the relationship between epigenetic changes in the development and progression of HPV associated with cervical cancer. A systematic literature search was conducted in major databases using predefined inclusion and exclusion criteria. Studies that investigated the expression, function, and clinical significance of miRNAs, DNA methylation, and the expression of oncoproteins in HPV-related cervical cancer were included. Data extraction, quality assessment, and synthesis were performed to provide a comprehensive overview of the current state of knowledge. We provide an overview of the studies investigating miRNA expression in relation to cervical cancer progression, highlighting their common outcomes and their weaknesses/strengths. To achieve this, we systematically searched the Pubmed database for all articles published between January 2018 and December 2023. Our systematic review revealed a substantial body of evidence supporting the pivotal role of miRNA dysregulation in the pathogenesis of HPV-related cervical cancer and related oncoproteins. From the 28 studies retrieved, miR-124, FAM194/miR-124-2, and DNA methylation are the most frequently down- or up-regulated in CC progression. Notably, FAM194/miR-124-2 and DNA methylation emerged as a promising molecular marker for distinguishing between cases requiring immediate surgical intervention and those amenable to a more conservative wait-and-see approach. This systematic review underscores the critical involvement of microRNA in the context of HPV-related cervical cancer and sheds light on the potential clinical utility of FAM194/miR-124-2 and DNA methylation as a discriminatory tool for guiding treatment decisions. The identification of patients who may benefit from early surgical intervention versus those suitable for observation has important implications for personalized and targeted management strategies in the era of precision medicine.

## 1. Introduction

Cervical cancer (CC) is a significant health problem worldwide, and infection with high-risk HPVs is recognized as the main etiological factor [1]. Understanding the role of miRNAs and DNA methylation in HPV-related CC is crucial for developing new therapeutic strategies [2]. HPV is the primary cause of cervical cancer (CC) [1,2]. About 604,127 new CC cases are diagnosed annually worldwide, with about 342,000 deaths each year. CC represents the world’s fourth leading cancer in women and the second most common cancer in women aged 15 to 44 years [3], and in Italy, there were an estimated 2400 new cases in 2020 [4]. CC is preventable through HPV vaccination and screening, and it is treatable if diagnosed early and managed properly.

HPV is a double-stranded circular DNA virus containing the early region’s genes, which code for regulatory proteins (E1–E7) representing the main oncoproteins of HPV. During a persistent cervical infection, HPV E6 and E7 can promote DNA damage via p53 and pRb inactivation, resulting in impaired DNA checkpoint controls, leading to the subsequent transformation into cancer cells [5].

To date, more than 225 HPV types have been described [6]. According to their ability to cause precancerous lesions, these are categorized as low-risk (LR) and high-risk (HR) HPV types [7]. The HPV genotypes 16 and 18 are related to approximately 72% of all CC worldwide, and types 31, 33, 45, 52, and 58 cause a further 17% [8].

More than 90% of infected cases resolve spontaneously, with viral clearance within two years; however, the persistence of the HR-HPV infection can lead to low- (CIN 1) and high-grade (CIN3) cervical dysplasia with a subsequent progression risk to invasive CC [7]. Premalignant changes are reflected in a spectrum of histological abnormalities ranging from cervical intraepithelial neoplasia grade 1 (CIN 1) or mild dysplasia to moderate dysplasia (CIN 2) and severe dysplasia or carcinoma in situ (CIN 3) [9]. It is estimated that the risk of high-risk HPV infections is about 80%, but most infections are naturally eliminated by the host’s immune system [9,10]. Only lesions caused by the persistent infection of high-risk genotypes can develop into invasive cancer. The progression rates for CIN2 and CIN3 vary significantly, with different studies showing that CIN2 has a progression rate to CC of approximately 5–10%, while CIN3 has a higher progression rate of around 10–20%. The regression rates for CIN2 are reported to be between 40 and 60%, and for CIN3, around 30–40%. According to the literature, among CIN2 patients who were not receiving treatment, the risk of regression surpassed 60%. The results for non-pregnant CIN2 patients who received conservative treatment at various follow-up intervals were compiled in a meta-analysis that included 36 trials [11]. At 24-month intervals, 334 of 1257 women (32%) continued to have CIN2, 819 of 1470 untreated CIN2 women (50%) reverted to CIN1 or less (CIN1−), and 282 of 1445 women (18%) advanced to CIN3 or worse (CIN3+); the corresponding rates for patients under 30 were 60%, 29%, and 11%, respectively. The natural course of CIN2/3 in expectant mothers was similarly calculated in another meta-analysis: CIN2 lesions had pooled rates of regression of 59%, with 40% for persistence, and only 1% for advancement in the subgroup analysis [12].

The many existent HPV subtypes in low- or high-risk HPVs (hr-HPV) are based on their oncogenic potential. Low-risk HPVs (like HPV6, 11, 42, 43, and 44) are commonly responsible for benign epithelial lesions (verrucae, warts, and papillomas), while high-risk HPVs (like HPV16, 18, 31, 33, 34, 35, 39, 45, 51, 52, 56, 58, 59, 66, 68, and 70) are linked to the development of cancer (cervical, anal, penile, vulval, vaginal, and oropharyngeal) [13,14]. As for cervical cancer, about 70% of invasive neoplasms are caused by HPV16 and HPV18 genotypes [15]. About 5–10% of low-grade lesions evolve to high-grade lesions, while most (about 60%) resolve spontaneously. Premalignant changes are reflected in a spectrum of histological abnormalities ranging from cervical intraepithelial neoplasia grade 1 (CIN 1) or mild dysplasia to moderate dysplasia (CIN 2) and severe dysplasia or carcinoma in situ (CIN 3) [9]. It is estimated that the prevalence of high-risk HPV infections is about 80%, but most infections are naturally eliminated by the host’s immune system [9,10], and only lesions caused by the persistent infection of high-risk genotypes can develop into invasive cancer. Once a high-grade lesion develops, the percentage of those that spontaneously regress is about 33%, while 10% progress to invasive cervical cancer [16]. Overall, HPV infection is not sufficient to induce malignant CC transformation, which also implies other multiple pathogenic factors, such as viral integration status, virus genotype, viral load, and genetic and epigenetic modulation. In recent years, the molecular understanding of malignant transformation and epidemiologic information related to HPV has led to the development of many strategies based on new biomarkers aimed at better detection and early intervention in patients with cervical dysplasia and CC [15,16].

The dysregulation of oncoproteins E6 and E7 can lead to the development of cancer, and the E6 protein causes the loss of activity of p53 by promoting its degradation, while E7 causes the loss of cell cycle control by binding to the cyclin-dependent kinase inhibitor [13]. In addition, viral oncoproteins can proceed to interfere with epigenetic mechanisms and modify DNA methylation, histone methylation, and transcription [16]. Many studies indicate that microRNA deregulation contributes to cervical cancer tumorigenesis [17]. MiRNA alterations drive the progression of cervical cancer from CIN 1 to full blown cancer [18]. The importance of miRNAs in cervical tumors is linked to the fact that the miRNA loci are associated with fragile sites, known as insertion sites of the HPV virus in cervical tumors [16,18]. MiRNAs and DNA methylation are two relevant epigenetic cell modifications that have emerged in recent years as the most critical players in the regulation of gene expression [19]. MiRNAs are short non-coding RNAs that can regulate the expression of several target genes via complementary binding to specific seed sequences [20,21]. Considering that the seed sequence can be formed by 2–8 nucleotides and that the complementarity may also be imperfect, a single miRNA may potentially modulate hundreds of mRNAs [21]. Over time, the role of miRNAs has been progressively clarified, and even if some aspects have not yet been completely understood, miRNAs may represent biomarkers or surrogate markers of diagnosis and prognosis [21]. These small RNAs are commonly encoded by viruses that undergo long-term persistent infection, including HPV, and can act as oncogenes to promote carcinogenesis, or as tumor suppressors targeting and neutralizing oncogene messenger-RNA [22]. MiRNA deregulation contributes to CC tumorigenesis, driving its progression from CIN1 to invasive cancer. Furthermore, the genes encoded by the virus can influence the miRNA expression in cervical cells [22]. The exosomes have emerged as a novel source of non-invasive tumor biomarkers. The unique bilayer membrane structure of exosomes offers protection against external RNases and proteases, leading to enhanced stability of the enclosed mRNAs, miRNAs, and functional proteins, thus making exosomes highly sensitive markers for disease diagnosis. The cargo in tumor-derived exosomes, such as the range of miRNAs, can also serve as biomarkers, offering valuable targets for early detection, diagnosis and treatment [23]. Thus, miRNAs with 19 to 24 nucleotides play a key role in tumor initiation, progression, and dissemination through influencing continued proliferative capacity, resistance to apoptosis, induction of invasion and metastasis, increased angiogenesis, and avoidance of growth inhibition signals [15]. Therefore, miRNAs are potential candidates in oncology as diagnostic biomarkers, prognostic biomarkers, therapeutic targets, and preventive screening programs [22]. MiRNA deregulation contributes to CC tumorigenesis, driving its progression from CIN1 to invasive cancer. Furthermore, the genes encoded by the virus can influence the miRNA expression in cervical cells [21]. MiRNAs can regulate both tumor suppressor genes and oncogenes, and the altered expression of miRNAs represents an early event in the induction of carcinogenesis through HPV infection [2,24]. In CC, the expression of some miRNAs increases (miR-20a, miR-20b, miR-93, miR-224) while that of others decreases (miR-127, miR-143/145, miR-218) [25,26,27]. Genomic DNA methylation has been proposed as an additional marker to increase the sensitivity and predictivity in the detection of cervical dysplasia [28,29]. The methylation markers CADM1, FAM19A4, and MAL could be high-performance markers for CC screening [2,25,30,31]; additionally, long interspersed nuclear elements-1 (LINEs-1) methylation has been proposed as a surrogate marker to assess total DNA methylation levels in cancer tissues and blood samples [32,33,34]. The purpose of this systematic review was to identify, in the most recent scientific literature, one or more epigenetic markers that can be implemented to currently available screening for cervical cancer, with a particular focus on miRNAs, DNA methylation, and oncoprotein expression. MiRNA changes and the degree of DNA methylation were then compared. Six studies focused on the analysis of oncoproteins involved in the carcinogenic process (particularly p16, E4, and E7). Many of the analyzed studies focused on the greater correlation between cancerous lesions and miRNA modulation (especially mir124-2), rather than the infection of the specific HPV genotypes (high-risk HPV species, 16 and 18) [34]. The authors of those studies report the high variability of miRNA expression, especially among normal samples, and identify miRNAs that are significantly up- or down-regulated in pre- or malignant vs. normal samples. The integration of HPV DNA into the host genome can lead to alterations in DNA methylation patterns and miRNA expression, contributing to the progression from normal epithelium to cervical intraepithelial neoplasia and ultimately to cervical cancer. This review aims to examine the relationship between epigenetic changes in the development and progression of HPV associated with cervical cancer, providing an up-to-date overview of the involvement of miRNAs and DNA methylation in the pathogenesis of HPV-related CC.

## 2. Results

Twenty-eight studies were finally identified and included in this review (Figure 1).

An overview of the 28 identified studies is provided in Table 1, which reports the PICO (participants, interventions, comparisons, outcomes) patient eligibility criteria for each one.

The methodological quality was evaluated using the Quality Assessment of Comparative Diagnostic Accuracy Studies (QUADAS-2) checklist, it reported in Table 2. The quality assessment of selected studies was performed according to the adapted Newcastle-Ottawa Scale. This is based on (1) clarity of the study objective, (2) sample selection (representativeness of the sample; sample size; response rate9; (3) comparability (comparability of participants from different outcome groups), (4) outcome (assessment; statistical tests).

An overview of the main findings related to biomarker (miRNA, methylation) analyses, as inferred from the 28 identified studies, is reported in Table 3. The techniques most frequently used to evaluate miRNA expression were multiplex real-time and DNA methylation-specific PCR (19 articles) and quantitative real-time PCR (qPCR, 7 articles). In most of the studies reviewed (11 articles), samples of normal cervical tissues were analyzed in comparison with tissues from precancerous, mostly CIN2/3 (20 articles), and cancerous lesions (18 articles) (Figure 2). Most studies analyzed formalin-fixed tissue (one article), formalin-fixed paraffin-embedded tissue (FFPE) or frozen (later stored in RNA or PBS) (three articles), and air-dried cytological cervical tissues samples (one article). All specimens were collected at enrollment, before any treatment. Most of the studies based their research on cytologic and histologic samples, while regarding the HPV genotypes analyzed most of the articles focused on HPV16 and HPV18 (Figure 3).

### 2.1. Epigenetic Changes and Developmental Progression of HPV Disease Associated with Cervical Cancer

Stable, long-term alterations in DNA, known as epigenetic modifications, are a normal evolutionary process that results in essential environmental adaptations. Although they do not alter the DNA sequence, they do have an impact on genomic stability and gene expression. Epigenetic modifications are significant for several biological functions [2]. Through the activation of oncogenes, silencing of tumor suppressor genes, and aggravation of abnormalities in DNA repair pathways, epigenetic alterations may play a crucial role in cancer cells. While a persistent, high-risk HPV infection is directly linked to cervical cancer, there are several epigenetic modifications that have been found in the viral DNA and the infected cells’ genomes. These include histone modification, DNA methylation, and gene silencing by non-coding RNAs, which both start and maintain epigenetic changes [20,21]. While a chronic high-risk HPV infection is directly linked to CC, the viral DNA and genome of infected cells have been found to exhibit several epigenetic modifications. There have been reports of hypomethylation of the whole DNA, hypermethylation of host cell tumor suppressor genes, and hypermethylation of the E2 binding site found in the LCR of the viral genome. The carcinogenic process also involves alterations in ncRNA expression patterns, histone modifications, and acetylation [25]. In this study, we discuss some of these epigenetic pathways involved in the beginning and development of CC.

### 2.2. HPV and DNA Methylation

DNA methylation is an epigenetic mechanism that involves the transfer of a methyl group on the C5 position of cytosine to form 5-methylcytosine. This mechanism regulates gene expression in two ways: by recruiting proteins involved in gene repression and by inhibiting the binding of transcription factors to DNA. The DNA methylation pattern in the genome changes during development because of a process involving both DNA methylation and demethylation, giving differentiated cells a DNA methylation pattern specific to the regulation of gene transcription in that tissue [34]. During our study, we identified four articles analyzing methylation levels of genes indicated as potential biomarkers.

According to Rogeri, C. et al., the hypermethylation of the hsa-miR-124-2, SOX1, TERT, and LMX1A genes could be a promising biomarker for cervical cancer precursor lesions, independent of hr-HPV status, and can be performed with samples collected for cervical cytology and HPV DNA testing. Indeed, they found that increased methylation of hsa-miR-124-2 directly correlated with the decreased expression of miR-124, implying that methylation is functionally involved in cervical carcinogenesis [35].

In their study, Liu, J et al. suggested KDM5A (lysine-specific demethylase 5A) as a possible biomarker of progression to carcinoma. Indeed, they discovered that KDM5A-induced E7 oncoprotein upregulation promoted the proliferation and invasiveness of cervical cancer cells in vitro and in vivo, which was associated with a poor prognosis in cervical cancer patients [36]. Ou, R. et al., on the other hand, proposed lysine-specific demethylase 2A (KDM2A) as a potential medical management biomarker in cervical cancer. KDM2A upregulation is induced by HPV16 E7 and promotes the proliferation and invasion of cervical cancer cells, indicating a poor prognosis. KDM2A physically interacts with the tumor suppressor miR-132 promoter and inhibits its expression, effectively acting as a tumor activator [37]. In their study, Fullár A. et al. focused on TFPI-2, a gene found to be hypermethylated only in cancer cells. TFPI-2 can be considered a tumor suppressor since it plays a role in the suppression of invasiveness by cervical cancer. The inactivation of the TFPI-2 gene has been demonstrated using two well-known epigenetic regulatory mechanisms in cervical cancer cells and tumor-associated cervical fibroblasts [38].

According to Vink J. et al., E4 expression dramatically dropped from CIN2 to CIN3 and with a rising score of immunohistochemical expression of p16 and Ki-67 (referred to as “immunoscore”). Methylation positivity and miR124-2 expression increased significantly from CIN2 to CIN3 (*p* < 0.001). Although P16INK4A expression is not predictive of clinical behavior or the prognosis of CIN lesions, it has been linked to the severity of the CIN grade [39]. According to Zummeren, M. et al., extensive E4 expression decreased with increasing CIN grade and immunoscore; it was missing in carcinomas and most common in lesions with cumulative immunoscores of 1–3 or traditionally rated CIN1 [40]. According to Leeman, A. et al., E4-positive staining decreases as SIL/CIN grade increases. Lesions with diffuse but limited E4- and p16-positive staining are likely to be early transforming and productive lesions and may have a higher likelihood of spontaneous regression, whereas patients with methylation-positive cervical cytology specimens and largely p16-positive, E4-negative lesions are more likely to have ≥HSIL/CIN3 lesions [41].

According to Del Pino, M. et al., HSIL/CIN2-3 and CC lesions have significantly higher levels of CADM1 and MAL methylation than normal and LSIL lesions. It was observed that the methylation rate of CADM1, MAL, and miR124 increases with the severity of the lesion [42]. In the article by Jiao, X. et al., it is therefore configured as a potential biomarker, since SEPT9 hypermethylation exhibits high sensitivity and specificity in the diagnosis of cervical cancer. They also found that the development of radiosensitivity is partially caused by the suppression of SEPT9 expression [43].

A member of the lysyl oxidase (LOX) family, lysyl oxidase-like 2 (LOXL2), is essential for catalyzing the synthesis of collagen and elastin cross-links in the extracellular matrix (ECM). It is a protein found in numerous types of cancer, in which it has been linked to the development of cancer cell proliferation, invasion, metastasis, and angiogenesis, and its expression has been suggested to be linked to poorer prognosis. In cancer tissue, Cao C. et al. discovered that LOXL2 is highly expressed. High levels of LOXL2 expression were linked to worse overall survival (OS) and disease-free survival (DFS) in cervical carcinoma. The authors then investigated the relationship between LOXL2 expression and LOXL2 promoter DNA methylation, finding that 14 CpG islands of LOXL2 were significantly and negatively correlated with LOXL2 gene expression in cervical cancer [44].

### 2.3. HPV and miRNAs

MiRNAs are small non-coding RNAs capable of influencing messenger RNA (mRNA) through recognition sites in the 3′untranslated region (UTR, responsible for regulating its stability), altering its gene expression levels. Specifically, miRNAs can induce suppression or increase cellular competition for miRNA binding sites by interacting with long non-coding RNAs (lncRNAs), circular RNAs (circRNAs), and pseudogenes. It follows that the expression levels of miRNAs play an important role in carcinogenesis and other diseases [59]. Different articles analyzing miRNAs involved in cervical cancer studies along with the methodologies are given in Table 4.

## 3. Discussion

In summary, our review highlights the significant findings from host DNA studies, which demonstrate the crucial role of DNA methylation and miRNA expression in the regulation of key oncogenes and tumor suppressor genes in HPV-related cervical carcinogenesis. These alterations in the host genome contribute to the progression and persistence of the disease. Additionally, HPV DNA studies reveal the mechanisms by which HPV integrates into the host genome, leading to genetic instability and further promoting carcinogenesis. Understanding these molecular interactions provides valuable insights into potential biomarkers for early detection and targets for therapeutic intervention in HPV-related cervical cancer.

The FAM19A4/miR124-2 methylation diagnostic test can detect cervical cancers with high accuracy (about 98%), particularly advanced CIN lesions [15,61,62]. These are CIN2/3 lesions detected in patients with long-standing HPV infection and show a methylation profile like that found in cancer. Based on these parameters, they are considered to have a high risk of progression in the short term [59]; therefore, among future screening algorithms, this study emphasizes the importance of also considering the age of patients [59].

A study by De Strooper, L. et al. analyzed the FAM19A4 and miR124-2 DNA methylation derived from PAP-test samples, demonstrating good clinical performance in detecting cervical cancer and advanced CIN lesions of the HPV-positive women. They demonstrated that the absence of FAM19A4/miR124-2 methylation correlates with a low risk of cervical cancer in HPV-positive women aged 30 years and older [45].

An FAM19A4/miR124-2 methylation analysis, coupled or not with HPV16/18 genotyping, may be regarded as an objective alternative to cytology for triage testing of HPV-positive women in cervical cancer screening. This demonstrates that among HPV-positive women aged more than 30 years, a negative FAM19A4/miR124-2 methylation test provides comparable safety in terms of long-term CIN3+ risk versus a negative cytology test, while a positive methylation test justifies an immediate colposcopy referral [46]. A negative FAM19A4/miR124-2 methylation test result can exclude the presence of cervical carcinoma [28].

In the same year, Kremer, W.W. et al. investigated a potential role of methylation analysis in the cervical screening of women living with HIV in South Africa. They studied strategies that provided only a nonsignificant increase in sensitivity for CIN3 (67.8% as a triage test of women with ASC-US or LSIL cytology, 62.7% as a primary screening test with cytology triage of methylation positives) compared with cytology alone (59.3%), while the specificity decreased (from 91.6% for cytology alone to 85.0% and 87.2%, respectively). They also highlighted the problem of the costs of these strategies, considering them as non-effective in their case [47].

Positive lesions only for p16 and E4 are likely to represent lesions in the early transformation phase and have a higher probability of spontaneous regression. A loss of E4 expression in the worst lesion, on the other hand, is associated with methylation of FAM19A4/miR124-2 [41].

When FAM19A4/miR124-2 methylation analysis was combined with cytology, the CIN3 sensitivity was 84.6% (95% CI 78.3–90.8) and the specificity was 69.6% (95% CI 66.5–72.7). The cervical cancer incidence was significantly lower in methylation-negative women than in cytology-negative women over a 14-year period (risk difference 0.98%, 95% CI 0.26–2.0) [48].

A large retrospective multicenter clinical performance study demonstrated that the triage of HPV-positive women with the FAM19A4/miR124-2 methylation test provides objective and reproducible results in terms of the detection of ≥CIN3 (sensitivity for cervical cancer detection of 95% and 77.2% for CIN3, specificity of 78%) in four European countries [49]. Two further studies found the clinical regression (85%) of CIN2 and CIN3 in women with ASC-US/LSIL or an HPV16-negative status with a negative baseline FAM19A4/miR124-2 methylation test; a methylation test in conjunction with cytology or HPV genotyping was used to support a wait-and-see approach in women with CIN2/3 [50,51].

FAM19A4/miR124-2 methylation screening can improve the care of pregnant women with CIN3 by reducing overtreatment with no residual risk of progressive illness. A methylation-negative test can rule out progressive CIN3 and cancer (LR: 0, 95% CI: 0–0.203), indicating that these pregnant women can be safely maintained with conservative follow-up till after delivery [51].

EPB41L3, HPV16L1, HPV16L2, HPV18L2, HPV31L1, and HPV33L2 show high potential as prognostic biomarkers to identify progressive CIN2 [52].

According to the authors of two studies, the use of the FAM19A4/miR124-2 methylation test for advanced CIN2/3 lesions in young women makes this test an extremely effective and more specific tool to guide physicians in the management of women with CIN2/3 lesions [53,63].

Increased numbers of methylated cells and larger, genetically abnormal lesions may be associated with high methylation levels, which are symptomatic of increasing CIN disease [54].

Other studies investigated one or few miRNAs identified as dysregulated in CC progression.

Zummeren, M. et al. showed that a gradual transition of productive CIN to advanced transformative CIN and cancer can be seen depending on the expression levels of E4 and miR-124-2 expression. In particular, the authors point out that the exposure of the two is inversely proportional, with the expression of miR-124-2 increasing as the lesion grade increases [40]. Instead, the combination of CADM1 promoter methylation status, MAL, and miR124 alone can be considered a reliable biomarker to detect transforming HSIL/CIN lesions. The authors demonstrated that an increase in lesion severity also sees an increase in the methylation rate of CADM1, MAL, and miR-124.

In 2018, Babion I. et al. examined the expression levels of eight potential candidate miRNAs in cervical tissue samples, identifying five whose levels were significantly different between controls and women with CIN3 lesions. Using a logistic regression analysis, the authors demonstrated that the sensitivity and specificity for CIN3 detection obtained via a two-miRNA classifier (miR-15b-5p and miR-375) could be increased to 63 and 77%, respectively, including HPV16/18 genotyping [55]. The implementation of the routine diagnostics of HPV-related cervical dysplasia by investigating new molecular and epigenetic diagnostic tools achieves personalized prevention of CC, according to a value-based healthcare approach. The integration of personalized medicine into prevention may benefit citizens, patients, healthcare professionals, healthcare authorities, and industry, and will ultimately seek to contribute to better health and quality of life for Europe’s citizens [56,60].

Real-time PCR was used to estimate the levels of 25 miRNAs and 12 mRNAs involved in cervical carcinogenesis in 327 air-dried Papanicolaou-stained cervical smears from patients with cervical precancerous lesions, cancer, or no disease. It was shown that miR-375, miR-20, miR-96, CDKN2A, TSP4, and ECM1 can predict high-level lesions with a diagnostic sensitivity of 89.0% and a specificity of 84.2%. Furthermore, the methylation levels of mir124-2 and MAL promoters, HR-HPV genotypes, and viral load were studied in a subsample of the same competitors. The risk of high-grade lesions, as predicted by the classifier, is related to the frequency of methylation of MAL and mir124-2, but not to the HR-HPV genotype or viral load [56,57].

Other miRNAs have been proposed as biomarkers in identifying cancerous lesions of the cervix. miR-9-5p exhibits histotype-dependent expression, meaning that it is more prevalent in SCC and less prevalent in AC. This study suggests that the methylation of miR-9-1, one of its precursor genes, is associated with the low levels of miR-9-5p in ACs. Moreover, it was discovered that miR-9-5p is more expressed in SCCs and HPV16-positive cells and is dependent on both the histotype and hrHPV type. In conclusion, it was found that miR-9-5p exhibits oncomiR activity in cells derived from SCC, while it has a tumor suppressor function in cells derived from AC [20,63].

MiR-10b expression decreased in CC tissues compared with normal tissues, and that lower miR-10b expression was associated with vascular invasion, larger tumors, and HPV positivity [60].

MiR-362-3p, when over-expressed, correlated with a higher probability of survival in patients with SCC (HR: 0.561, 95% CI: 0.354–0.889, *p* = 0.014). The expression of miR-362-3p should be considered a reliable prognostic biomarker in cervical SCC, but not in ADC [58]. However, the detection of viral nucleic acid does not provide any information about the biological result of the interaction between HPV and the human host [21]. This piece of information is provided by an miRNA analysis in cervical tissue. This finding supports the combined use of HPV16/18 genotyping and microRNA detection as a triage test for HPV-positive women to identify subjects at high risk for cancer progression. The characterization of the CIN via multiple methods (HPV genotyping, miRNA, IHC, DNA methylation) is a new tool to identify subject at high risk for cancer evolution in advance. This would lead to the targeted treatment of those patients with a CIN at high risk of progression to invasive forms, thus personalizing both therapeutic and follow-up protocols. Indeed, the use of multiple biomarkers is a reasonable strategy to increase the predictivity of the performed analysis in identifying at an individual level the risk of CIN progression. MiRNAs and DNA methylation are two relevant epigenetic cell modifications that have emerged in recent years as the most critical players in the regulation of gene expression [20]. The methylation rate of CADM1, MAL, and miR-124 increases with the severity of the lesion [42]. It is well established that HPV16 and 18 infections dramatically increase the risk of CIN3 and cancer onset. Furthermore, performing this analysis before invasive diagnostic interventions facilitates the targeted treatment of only those lesions with a real risk of neoplastic progression, with important relapses for the patients, who would undergo invasive investigations only if strictly necessary.

The reviewed studies are characterized by a high heterogeneity. Differences exist between the techniques used to measure miRNA expression, although PCR has been the most common. A similar observation may be made for the analysis of DNA methylation, although pyrosequencing has been the most common technique. These methodological differences may contribute to the lack of full homogeneity in the findings obtained.

The characterization of novel epigenetic and other biological markers of HPV-related dysplasia could improve early diagnosis, our understanding of the risk of progression to invasive CC, and the clinical management of patients with the diagnosis of CIN I-II-III. The investigation of miRNA levels in cervical exfoliated cells certainly opens new possibilities for studying molecular markers in the context of screening programs. However, this systematic review has several limitations. First, the variation in sample sizes across selected studies may have affected the results of some miRNAs. Some studies have disproportionate sample sizes in each category, which may have impacted the precision of the data. However, most of the included studies had sufficient sample sizes and data on miRNA expression levels to draw valid conclusions.

## 4. Materials and Methods

### 4.1. Literature Search

This review was conducted following the Preferred Reporting Items for Systematic Reviews and Meta-Analysis Protocol (PRISMA) guidelines [33]. The literature search was conducted on PubMed using the two following search strings (terms searched in advanced search in “All Fields”): “HPV” AND “miRNA” AND “methylation” AND “cervical cancer”; “HPV” AND “microRNA” AND “methylation” AND “cervical cancer” as keywords.

The correlation between miRNA, DNA methylation, and HPV-related disease (CIN1, CIN2, CIN3, and CC lesions) was the main criterion of inclusion for this review.

A total of 125 studies were first identified. Before screening, 56 studies were removed because they were duplicates, leaving a total of 69 remaining studies to be screened. Of these, 13 were excluded: 1 was written in Czech, 7 were reviews, 1 was retracted, and 1 was not finished. A total of 56 results were assessed for eligibility: 5 were not pertinent to the subject of the research, and 23 were published before 2018. Studies focusing on cancers unrelated to cervical cancer were also excluded.

Twenty-eight studies were finally identified and included in this review (Figure 1).

The search was limited to articles published between 2018 and 2023 to maintain relevance. We cannot overlook that some important studies published before 2018 that might also have added value to the topic were not included in our analysis.

### 4.2. Literature Search Samples, Grading Lesion, miRNA, DNA-Methylation, and HPV Genotype Analysis

The studies included were mainly conducted on human material (26 items of 28): cervical scrapes (7 items), cervical biopsy (5 items), cytology (4 items), colposcopy (3 items), cervical tissues (2 items), surgical specimens (2 items), curettage (1 item), trachelectomy, hysterectomy, and pelvic lymph node dissection (1 item), and liquid-based cytology (1 item). MiRNAs, the target genes analyzed, and their functioning were collected as follows: miR-124-2 (4 items); CADM1, MAL, and miR124 promoter genes (2 items); FAM19A4/miR-124-2 (11 items); miR-10b (1 item); miR-9-5p, miR-15b-5p, miR-28-5p miR-100-5p, miR-125b-5p, miR-149-5p, miR-203a-3p, and miR-375 (2 items); miR-362-3p (1 item) (Table 4).

Lesion grades CIN1 (9 items), CIN2-3 (20 items), CC (18 items), and their comparison to control groups (11 items) were also considered (Figure 2), as well as HPV genotypes (HPV 16, 18: 57% items; HPV 16: 15% items; other HPV genotypes: 3,8% items; unspecified HPV genotype: 23% items) (Figure 3).

### 4.3. Quality and Reporting Appraisal

Two independent experts investigated the retrieved literature (G.C., P.A.). In case of disagreement between them, two others (AI, AP) were involved to reach a final consensus. No automation tools were used for this review.

The methodological quality was evaluated using the Quality Assessment of Comparative Diagnostic Accuracy Studies (QUADAS-2) checklist, as reported in Table 1. The quality assessment of selected studies was performed according to the adapted Newcastle–Ottawa Scale. This is based on the (1) clarity of the study objective, (2) sample selection (representativeness of the sample; sample size; response rate), (3) comparability (comparability of participants from different outcome groups), and (4) outcome (assessment; statistical tests). Green color indicates a low risk of bias, red refers to a high risk of bias, and yellow indicates no information.

The data obtained from the full-text analysis were grouped into Microsoft Excel tables (Table 1, Table 2 and Table 3), based on which an analysis of the results was prepared. The eligibility criteria in accordance with PICO (participants, interventions, comparisons, outcomes) is reported in Table 1.

## 5. Future Research

The epigenetic aspects of the HPV genome reveal that different HPV genotypes may have different effects on the host epigenetic machinery. In HPV16/18, the host methylation machinery is triggered strongly to methylate the viral and host genome. The methylation of HPV16 and HPV18 has the potential to differentiate normal from ≥CIN1 cervical lesions. HPV infection, viral genotype, physical viral DNA state, and carcinogenic risk all affect changes to the methylation status of cellular DNA. These methylation biomarkers showed satisfactory clinical results in patients with CIN3 and more severe lesions. However, the goal of early detection is to improve the detection of precancerous lesions before the ≥CIN3 stage by, for instance, identifying new cervical cancer methylation biomarkers, possibly by including an analysis of the methylation of other genes or via HPV typing. MiRNA deregulation contributes to CC tumorigenesis, driving its progression from CIN1 to invasive cancer. Furthermore, the genes encoded by the virus can influence the miRNA expression in cervical cells [15]. MiRNAs can regulate both genes related to tumor suppressor genes and oncogenes, and the altered expression of miRNAs represents an early event in the induction of carcinogenesis by HPV infection [2,25]. The implementation of routine CC prevention by investigating new molecular and epigenetic diagnostic markers is demanding. This study aimed to evaluate the feasibility and utility of including epigenetic alterations such as miRNA regulation and DNA methylation in CC screening; elucidating the association between genetic and epigenetic modifications and the characterization of the lesion can improve early diagnosis, the understanding of neoplastic progression risk, and the clinical management of patients, with the aim of setting up personalized screening protocols for CC. Since miRNAs are linked to treatment resistance in cervical cancer, new combination therapies involving miRNA inhibitors, or their addition to chemotherapy or radiotherapy, may be created. These findings point to varieties of molecular targeted therapies and miRNA-specific customized therapies, and miRNAs are anticipated to play a significant role in the detection and management of CC.

## 6. Conclusions

Alterations to the methylation status of cellular DNA are influenced by HPV infection, the viral genotype, the physical state of the viral DNA, and oncogenic risk. The E6 and E7 oncoproteins of HPV 16 induce the overexpression of DNA methyltransferase enzymes, which can catalyze the aberrant methylation of protein-coding and miRNA genes that are susceptible to regulation via methylation. Increased knowledge of the molecular changes in cervical precancer and cancer, such as genetic and/or epigenetic changes associated with cervical neoplastic progression combined with HR-HPV infection, will be important for the next generation of screening programs. Indeed, the use of multiple biomarkers is a reasonable strategy to increase the predictivity of the performed analysis in identifying the risk of CIN progression at an individual level.

## Figures and Tables

**Figure 1 ijms-25-12714-f001:**
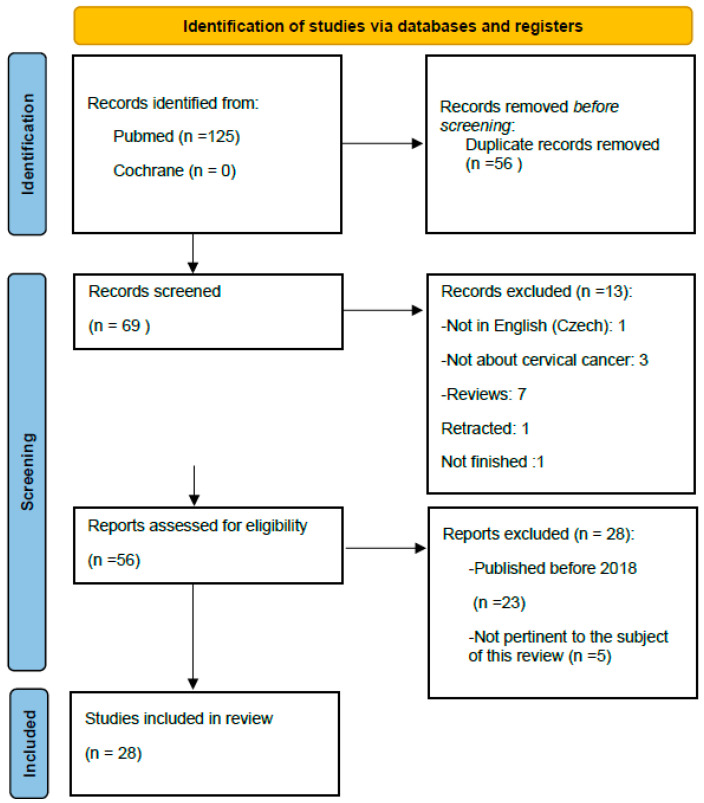
PRISMA flowchart for the studies included in the systematic review.

**Figure 2 ijms-25-12714-f002:**
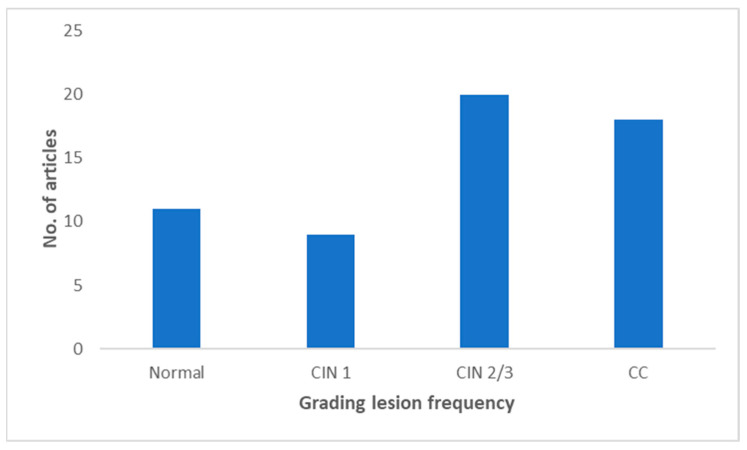
Grading lesion frequency based on the number of articles. The number of articles is shown on the *y*-axis, and the degree of the lesion (CIN1, CIN2/3, CC) and a comparison with the normal condition are shown on the *x*-axis. In most of the studies reviewed (8 articles), samples of CIN 1 lesions of cervical tissues were analyzed in comparison with tissues with high levels of CIN2/3 lesions (20 articles) and cancerous lesions (CC) (18 articles).

**Figure 3 ijms-25-12714-f003:**
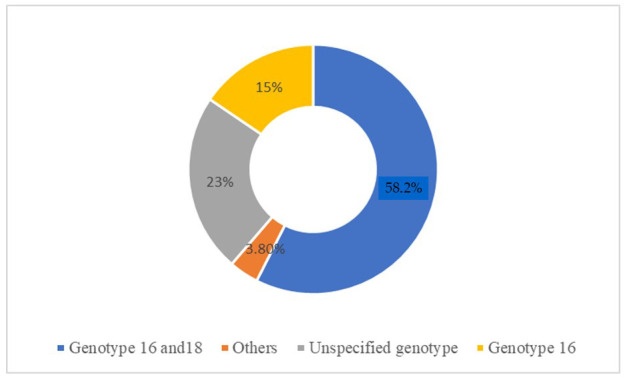
Most of the studies based their research on cytologic and histologic samples. The HPV genotypes analyzed are HPV16 and HPV18 in 57% of the studies (blue color) and only HPV 16 reported in 15% of the studies (yellow color), while 23% analyzed unspecified genotypes (gray color) and 3.80% analyzed other genotypes (orange color).

**Table 1 ijms-25-12714-t001:** Table for eligibility criteria in accordance with PICO elements (participants, interventions, comparisons, outcomes).

Reference	P (Patient/Population)	I (Intervention)	C (Comparison/Control)	O (Outcome)
[19]	CC patients (86)	Cervical Cancer Tissues 21, miR-30	miRNA microarray	Predicted miRNA-mRNA interaction (miR-9-1 hyper-methylation
[31]	CC patients (*n* = 519).	Cervical scrapes	Evaluated FAM19A4/miR124-2 methylation analysis in a large, worldwide series of cervical cancer.	Methylation analysis of FAM19A4 and miR124-2 genes of high-risk (hr) HPV-positive women
[35]	CC patients (*n* = 447)	Colposcopy	(1) Cervices without cervical intraepithelial neoplasia; (2) cervices with a CIN grade of 1 (CIN 1); and (3) cervices with a CIN grade of 2 or 3 (CIN 2/3).	Methylation pattern for a panel of 15 genes was analyzed via quantitative methylation-specific PCR (qMSP) and compared between the groups (≤CIN 1 vs. CIN 2+).
[36]	CC patients (*n* = 42)	Cervical scrapes	In vivo metastasis assay.	Discovery of function of KDM5A in cervical cancer progression as a novel prognostic biomarker and target for the clinical management of this malignancy.
[37]	CC patients (*n* = 81)	Surgical specimens	Biomarker comparison in cancer vs. healthy tissue.	The function of KDM2A-miRNAs on cervical cancer was investigated in vitro and in vivo, and KDM2A was suggested as a new prognostic biomarker of cervical cancer.
[38]	*n*/a	Fresh tumorous and normal areas of surgical specimens	In vitro models of monocultures and co-cultures.	Characterization of the mechanism of *TFPI2* downregulation in tumor-associated fibroblasts and tumor cells. Inactivation of the *TFPI2* gene plays a strategic role in the progression of cervical cancer.
[39]	CC patients (*n* = 497)	Cervical scrapes	Higher E4 positivity in CIN2 lesions than in CIN3 lesions.	HPV status and FAM19A4/miR124-2 methylation analysis.
[40]	CC patients (*n* = 115)	Paraffin-embedded cervical biopsy	No dysplasia, CIN1, CIN2, CIN3, or cancer via classical histomorphological grading criteria or via an immunoscore.	HPV E4 expression and DNA hypermethylation of CADM1, MAL, and miR124-2 genes.
[41]	CC patients (*n* = 318)	Biopsy	Investigated the relationship between staining patterns of p16 and E4 IHC in the worst biopsy, and the relationship between these and FAM19A4/miR124-2 methylation status in cytology.	Expression of p16 and HPV E4 on biopsy samples and methylation of FAM19A4 and miR124-2 on cervical cytology samples.
[42]	CC patients (*n* = 131)	Biopsy	Biomarker comparison in cancer vs. healthy tissue.	Methylation analysis for CADM1, MAL, and miR124 alone, in combination, and in association with hrHPV genotyping for the detection of transforming HSIL/CIN in several well-characterized cervical biopsies.
[43]	CC patients (*n* = 80)	biopsy	Biomarker comparison in cancer vs. healthy tissue.	Identification of SEPT9 methylation as CC biomarker.
[44]	CC patients (*n* = 176)	Cervical Cancer Tissues	LOXL2 expression in vitro assays.	Identification of a new potential therapeutic target for cervical carcinoma via molecular analyses.
[45]	CC patients (*n* = 1040)	Cytology, and HPV16/18 genotyping of HPV-positive women	FAM19A4/miR124-2 methylation, cytology, and HPV16/18 genotyping of HPV-positive women.	Cumulative CIN3+ incidences over 3 screening rounds (5-year intervals) of 4 triage strategies were compared.
[46]	CC patients (*n* = 1025)	Colposcopy	FAM19A4/miR124-2 methylation, cytology.	A negative FAM19A4/miR124-2 methylation triage test provides a similar long-term CIN3+ risk compared with a negative cytology triage test. FAM19A4/miR124-2 methylation analysis was found to be a promising alternative to cytology for triage.
[47]	CC patients (*n* = 318)	Cytology screening and colposcopy-directed biopsy	Cytology, HPV-test, *FAM19A4/miR124-2* methylation.	Cytology provided highest specificity but lowest sensitivity, whereas a single HPV test provided the highest sensitivity but the lowest specificity. Combining cytology with methylation did not improve the performance compared with cytology alone.
[48]	CC patients (*n* = 979)	HPV-based cervical screening via cytology and methylation analysis	FAM19A4/miR124-2 methylation, cytology.	Evaluation of FAM19A4/miR124-2 methylation analysis as a triage test for HPV-positive women: cancer incidence was significantly lower for methylation-negative women compared to cytology-negative women.
[49]	CC patients (*n* = 2384)	Cervical scrapes	FAM19A4/miR124-2 methylation, cytology.	FAM19A4/miR124-2 methylation appears to be an alternative/supplement to cytology as a triage method to be investigated in real-life pilot implementation.
[50]	CC patients (*n* = 294).	Colposcopy	HPV-positive women with a borderline or mild dyskaryosis	FAM19A4/miR124-2 methylation, HPV16/18 genotyping, and HPV16/18/31/33/45 genotyping in HPV-positive women.
[51]	CC patients (*n* = 135)	biopsy	FAM19A4/miR124-2	FAM19A4/miR124-2 methylation Progressive CIN3 in pregnant women.
[52]	CC patients (*n* = 114)	Cervical screening	FAM19A4/miR124-2 methylation, HPV-test, cytology.	Most women with untreated CIN2/3 and a negative baseline *FAM19A4/miR124-2* methylation test showed clinical regression. Methylation, in combination with cytology or HPV genotyping, can be used to support a wait-and-see policy in women with CIN2/3.
[53]	CC patients (*n* = 1061)	Cervical curettage	Methylation positivity and CIN grade.	Evaluating the FAM19A4/miR124-2 methylation test.
[54]	CC patients (*n* = 106)	Liquid-based cytology (LBC) samples	Hypermethylation of the human genes FAM19A4 and miR124-2	Testing for methylation of FAM19A4/miR124-2 as a triage for HPV-positive women appears to be useful to identify women at risk of cancer development, especially adenocarcinoma
[55]	CC patients (*n* = 209)	Cervical scrapes.	Biomarker comparison in cancer vs. health tissue	Expression levels 10 candidates’ miRNA markers in the different biological groups CIN0/1, CIN2, and CIN3
[56]	CC patients (*n* = 283)	Cervical scrapes	Expression levels of the eight candidate miRNAs in cervical tissue samples (*n* = 58) and hrHPV-positive cervical scrapes from a screening population (*n* = 187) and cancer patients (*n* = 38)	Evaluated the clinical value of eight miRNAs (miR-9-5p, miR-15b-5p, miR-28-5p, miR-100-5p, miR-125b-5p, miR-149-5p, miR-203a-3p, and miR-375) on cervical scratches for triaging hrHPV-positive women in cervical screening
[57]	CC patients (*n* = 327)	Cervical scrapes	miR-375, miR-20, miR-96, mir124-2	Evaluated the clinical value of miR-20, miR-96, CDKN2A, TSP4, and ECM1, predicted high-grade lesions with diagnostic sensitivity of 89.0%, specificity of 84.2%
[58]	*n*/a	Oncogene	Feedback loops between miRNAs and DNA methylation in human cancers.	miRNA dysregulation and aberrant DNA methylation are involved in tumor initiation, progression, and metastasis.
[59]	CC patients (*n* = 529)	HPV screening followed by triage with cytology	Biomarker comparison in cancer vs. healthy tissue.	Full genotyping and FAM19A4/miR124-2 methylation analysis in high-risk human papillomavirus-positive samples from women over 30 years old participating in cervical cancer screening.
[60]	CC patients (*n* = 70).	Cervical tissues	miR-10b expression measured in 51 CC cases and 19 normal controls.	Less expression of miR 10b in CC associated with larger tumor, vascular invasion, and HPV type 16 positivity

**Table 2 ijms-25-12714-t002:** Quality Assessment of Comparative Diagnostic Accuracy Studies (QUADAS-2) checklist.

Studies	Risk of Bias										Concerns Regarding Applicability		
	Patient Selection			Index Text	Reference Standard			Flow and Timing			Patient Selection	Index	Reference Stand
	Was a Consecutive or Random Sample of Patients Enrolled?	Was a Case–Control Deign Avoided?	Did the Study Use Inappropriate Exclusion Criteria?	Were the Index Test Results Interpreted Without Knowledge of the Result of Reference Standard?	If a Threshold Was Used, Was It Pre-Specified?	Is the Reference Standard Likely to Correctly Classify the Target Condition?	Were Reference Standard Results Interpreted Without Knowledge of the Index Test?	Was There an Appropriate Interval Between the Index Test and the Reference Standard?	Did All Patients Receive the Same Reference Standard?	Were All Patients Included in the Analysis?	Is There Concern That Included Patients Do Not Match the Review Question?	Is There Concern That the Index Test, Its Conduct, or Its Interpretation Differ from the Review Question?	Is There Concern That the Target Condition as Defined by the Reference Standard Does Not Match the Review Question?
[19]	+	+	+	+	+	+	+	+	+	+	+	+	+
[31]	+	+	+	+	+	+	+	+	+	+	+	+	+
[35]	+	+	+	+	+	+	+	+	+	+	+	+	+
[36]	+	+	+	+	+	+	+	+	+	+	+	?	+
[37]	+	+	+	+	+	+	+	+	+	+	+	?	+
[38]	+	+	+	?	+	+	+	+	+	?	+	?	+
[39]	+	+	+	+	+	+	+	+	+	+	+	?	+
[40]	+	+	+	+	+	+	+	+	+	+	+	+	+
[41]	+	+	+	+	+	+	+	+	+	−	+	+	+
[42]	+	+	+	+	+	+	+	+	+	−	+	?	+
[43]	+	+	+	+	+	+	+	+	+	+	+	+	+
[44]	+	+	+	+	+	+	+	+	+	+	+	+	+
[45]	+	+	+	+	+	+	+	+	+	+	+	+	+
[46]	+	+	+	+	+	+	+	+	+	+	+	?	+
[47]	+	+	+	?	+	+	+	+	+	−	+	−	+
[48]	+	+	+	+	+	+	+	+	+	−	+	+	+
[49]	+	+	+	?	+	+	+	+	+	−	+	?	+
[50]	+	+	+	+	+	+	+	+	+	−	+	?	+
[51]	+	+	+	?	+	+	+	+	+	−	+	?	+
[52]	+	+	+	+	+	+	+	+	+	+	+	+	+
[53]	+	+	+	+	+	+	+	+	+	+	+	+	+
[54]	+	+	+	+	+	+	+	+	+	+	+	+	+
[55]	+	+	+	?	+	+	+	+	+	−	+	+	+
[56]	+	+	+	+	+	+	+	+	+	+	+	?	+
[57]	+	+	+	+	+	+	+	+	+	+	+	+	+
[58]	+	+	+	+	+	+	+	+	+	?	+	+	+
[59]	+	+	+	?	+	+	+	+	+	−	+	?	+
[60]	+	+	+	+	+	+	+	+	+	+	+	+	+

The symbol + indicates a low risk of bias, − refers to a high risk of bias, and ? indicates no information.

**Table 3 ijms-25-12714-t003:** Biomarkers and focus of biomarkers in the reviewed articles. Variations in biomarkers in terms of up- (↑) and downregulation (↓) refer to a comparison between cancer and normal tissues.

Reference	HPV Genotype	Protein(s) Analyzed	MiRNAs Analyzed	Number of Subjects	Proposed Biomarkers	Methylation	MiRNAs	Clinical Aspect
[19]	16, 18		miR-221, miR-30 miR-138	86	Predicted miRNA-mRNA interactions miR-221-3p_BRWD3, miR-221-3p_FOS, and miR-138-5p_PLXNB2	↑ in SSC; ↓ in AC (miR-9-1 hypermethylation)	↓ miR-221 (*p* < 0.05) ↑ miR-30 ↓ miR-138 (*p* < 0.05)	HPV-driven transformation, emphasizing PITX2’s role and confirming miRNA-mRNA interactions, in cervical cancer development.
[31]	16, 18, others		FAM19A4/miRNA 124-2	519	FAM19A4/miR124-2 methylation	↑ Methylation	↑ miR-124-2 (*p* < 0.0001)	Cervical cancer across various subgroups, suggesting its potential as a sensitive triage test in HPV-based screening.
[35]	16, 18, 31, 33, 35, 39, 45, 51, 52, 56, 58, 59, 66, 68	HIC1, APC, CADM1, CDH1, DAPK1, JAM3, EPB41L3, C130RF18, MAL, PAX1, NKX6-1	miR-124-2 and target genes SOX1, TERT, LMX1A, DAPK1, EPB41L3, HIC1	447	hsa-miR-124-2, SOX1, TERT, LMX1A hypermethylation	↑ Methylation	↓ miR-124-2 (*p* = 0.001) (OR = 5.1)	Detecting precursor lesions in cervical cancer.
[36]	16	E7	microRNA-424-5p and target gene KDM5A	42	KDM5A	↑ Gene Expression	↑ miR-424-5p (*p* < 0.05)	Cervical cancer and therapeutic target.
[37]	16	E7	miR-132 and target gene KDM2A	81	KDM2A	↑ Gene Expression	↑ miR-132 *p* < 0.05	Cervical cancer progression by suppressing miR-132.
[38]	16		miR-23a and target gene TFPI-2	34	TFPI-2 inactivation	↓ Expression	↓ miR-23a (*p* < 0.05)	Cervical cancer women with HPV 16 positive.
[39]	n/a	E4, p16, KI67	FAM19A4/miR124-2	497	E4, p16, Ki-67 expression; FAM19A4/miR124-2 methylation	↓ E4; ↑ p16, ↑ Methylation (FAM19A4/miR-124-2)	↑ miR-124-2 (*p* < 0.001)	High-grade CIN lesions.
[40]	16,18, 31, 33, 35, 39, 45, 51, 52, 53, 56, 58, 59, 66, 67, 70	P16INK4A, E4, CADM1, MAL	MIR124-2	115	E4 expression; CADM1, MAL, miR124-2 methylation	E4 ↓; ↑ methylation CADM1, MAL, miR124-2	↑ miR124-2 (*p* < 0.05)	Productive to advanced transforming cervical lesions.
[41]	16, 18, 31, 33, 35, 39, 45, 51, 52, 56, 58, 59, 66, 68.	E4, P16	FAM19A4/miRNA 124-2	318	E4, p16 expression; FAM19A4/miR124-2 methylation	↓ E4; ↑ p16; ↑ methylation FAM19A4/miR124-2	↑ miR124-2 (*p* < 0.001)	Cervical squamous intraepithelial lesions.
[42]	16, 18, others	CADM1, MAL	miR-124	131	CADM1, MAL, miR-124 methylation	↑ Methylation	↑ miR124 Sensitivity 70.2% Specificity 57.8%	High-grade cervical intraepithelial lesions.
[43]	n/a	SEPT9	miR-375	80	SEPT9 methylation	↑ Methylation	↑ MiR-375 (*p* < 0.001)	Cervical cancer tissues and para-carcinoma normal tissues from cervical cancer patients.
[44]	n/a	LOXL2		176	LOXL2	↑ Methylation		LOXL2 expression in cervical carcinoma correlates with poor survival via EMT.
[45]	n/a		FAM19A4/miRNA 124-2	1040	FAM19A4/miRNA 124-2 methylation	↑ Methylation		Cervical cancer in HPV-positive women aged 30 and older.
[46]	16, 18		FAM19A4/miR124-2	1025	FAM19A4/miR124-2 methylation	↑ Methylation		Long-term CIN3+ risk among HPV-positive women triaged with FAM19A4/miR124-2 methylation analysis is comparable to cytology.
[47]	16, 18, others		FAM19A4/miR124-2	318	FAM19A4/miR124-2 methylation	↑ Methylation		HIV-positive South African women compared to cytology.
[48]	n/a		FAM19A4/miR124-2	979	FAM19A4/miR124-2 methylation	↑ Methylation		Cytology for detecting high-grade cervical lesions in HPV-positive women.
[49]	n/a		FAM19A4/miR124-2	2384	FAM19A4/miR124-2 methylation	↑ Methylation		Cervical cancer and precancerous lesions in HPV-positive women during cervical cancer screening.
[50]	16, 18, 31, 33, 35, 39, 45, 51, 52, 56, 58, 59, 66, 68		FAM19A4/miR124-2	294	FAM19A4/miR124-2 methylation	↑ Methylation	↑ miR124-2 (*p* < 0.001)	Risk-stratified HPV-positive women with low-grade cytology, potentially reducing the need for direct colposcopy referrals.
[51]	n/a		FAM19A4/miR124-2	135	FAM19A4/miR124-2 methylation	↑ Methylation		Progressive CIN3 in pregnant women.
[52]	31, 33, 35, 39, 45, 51, 52, 56, 58, 59, 66,67, 68		FAM19A4/miR124-2	114	FAM19A4/miR124-2 methylation	↑ Methylation		Women with untreated CIN2/3 and negative FAM19A4/miR124-2 methylation have a higher likelihood of clinical regression.
[53]	16, 18	E4; P16/Ki-67	FAM19A4/miRNA 124-2	1061	FAM19A4/miRNA 124-2 methylation	↑ Methylation	↑ miRNA 124-2 (*p* = 0.003)	CIN2/3 lesions in women aged <30 years.
[54]	n/a		FAM19A4/miR124-2	106	FAM19A4/miR124-2 methylation	↑ Methylation		Cervical cytology samples may indicate risk of adenocarcinoma up to 8 years before diagnosis.
[55]	16	FAM19A4	miR-15b, miR-125b, miR-149, miR-203a, miR-375, let-7b, miR-20a, miR-31, miR-93, miR-222	209	FAM19A4 methylation/classifier (miR-149, miR-20a, miR-93) expression	↑ Methylation, expression (miR-149, miR-20a, miR-93)	↑ miR-149 (*p* < 0.05); ↑ miR-20a (*p* < 0.05); ↑ miR-93 (*p* < 0.05)	Cervical disease in hrHPV-positive women
[56]	16, 18, others		miR-15b-5, miR-375, miR-15b-5p, miR125-5p, miR-149-5p, miR-203a-3	283	miR-15b-5p, miR-125b-5p, miR-149-5p, miR-203a-3p and miR-375) obtained from hrHPV-positive cervical CIN3		↑ miR-15b-5, Sensitivity 56.1% specificity 62.0% ↓ miR-125 Sensitivity 72.7% specificity 47% (*p* = 0.067) ↓ miR-149-5p Sensitivity 84.3% specificity 28.8% ↓ miR-375 Sensitivity 52.9% specificity 62.1%	Cervical scrapes feasibly detect cervical cancer early, offering potential for triaging high-risk HPV-positive women in screening programs.
[57]	16, 18, 31, 33, 35, 39, 45, 51, 52, 56, 58, 59	CDKN2A, TSP4, ECM1, MAL	miR-375, miR-20, miR-96, mir-124-2	327	Classifier (miR-375, miR-20, miR-96, CDKN2A, TSP4, ECM1)	n/a	MIR124-2 methylation	High-grade cervical lesions and cancer.
[58]	16, 18, 31, 33,35, 39, 45, 51, 52, 56, 58, 59, 66, 68,		miR-362-3p, miR-362-5p	89	miR-362-3p	↓ Expression	↓ miR-362-3p HR: 0.561, 95%CI: 0.354-0.889, (*p* = 0.014).	Cervical squamous cell carcinoma, with no prognostic value observed in adenocarcinoma.
[59]	16, 18, 31, 33, 35, 39, 45, 51, 52, 56, 58, 59, 66, 68		FAM19A4/miRNA 124-2	529	FAM19A4/miRNA 124-2 methylation negative	↓ Methylation		HPV-positive women over 30 in cervical cancer screening.
[60]	16, 18		miR-10b	70	miR-10b	miR-10b	↓ miR-10b (*p* < 0.001)	HPV-positive cervical cancer by targeting Tiam1.

**Table 4 ijms-25-12714-t004:** MiRNAs and target genes analyzed and their functioning.

miRNAs	Methodology and Functions	Number of Subjects	References
miR-124-2	Methylation analysis of the promoter regions of FAM19A4 and miR124-2 in liquid-based cytology samples, and multiplex quantitative methylation-specific PCR.	167	[30]
		447	[35]
		115	[40]
		318	[41]
CADM1, MAL, and miR124 promoter genes	DNA methylation profiling using a bead-based microarray followed by validation with pyrosequencing, and multiplex quantitative methylation-specific PCR (qMSP).	131	[42]
FAM19A4/miR-124-2	Immunohistochemical analysis of P16INK4A, Ki-67, and HPV E4 expression, and DNA methylation analysis of FAM19A4/miR124-2 using quantitative methylation-specific PCR (qMSP).	497, 529	[39,59]
		1040, 1025	[45,46]
		318, 979	[47,48]
		2384, 294	[49,50]
		135, 114	[51,52]
		1061	[53]
miR-124-2	hypermethylation of gene and quantitative methylation-specific PCR (qMSP).	109	[54]
miR-9-5p, miR-15b-5p, miR-28-5p miR-100-5p, miR-125b-5p, miR-149-5p, miR-203a-3p, miR-375	Quantitative RT-PCR to measure miRNA expression levels in cervical tissue samples and hrHPV-positive cervical scrapes and quantitative methylation-specific PCR (qMSP) for DNA methylation analysis.	283	[55]
		209 hrHPV-positive scrapes of women with CIN2/CIN3	[56]
miR-362-3p	Methylation-specific PCR (MSP)	89	[58]
miR-10b	Quantitative RT-PCR to measure miR-10b expression, methylation sequencing to analyze CpG sites, cell proliferation, apoptosis, migration, and invasion assays, and Western blot to assay the expression of the target gene	70	[60]

## Data Availability

The datasets used and/or analyzed during this study are available from the corresponding author on reasonable request.

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
