# Peer review of "The Role of microRNA Expression and DNA Methylation in HPV-Related Cervical Cancer: A Systematic Review"

_ijms, 2024, doi:10.3390/ijms252312714_

Round 1
Reviewer 1 Report (New Reviewer)
Comments and Suggestions for Authors
The authors presented the review entitled “The role of miRNAs and methylation in HPV-related cervical cancer: a systematic review”.
I have suggestions.
Add the PICO acronym (participants, interventions, comparisons, outcomes) in method section, for example, a table for eligibility criteria in accordance with PICO (participants, interventions, comparisons, outcomes).
Authors’ description: From the 28 studies retrieved, miR-124 and FAM194/miR-124-2 methylation are the most frequently down- and up-regulated in CC progression, respectively. How The point of view quantitative demonstrated the quality of article research. Explanation necessary.
Some words in the title “miRNAs and methylation” are not suitable as miRNA is a substantial but methylation is an event.
Author Response
Reviewer 1
COMMENT 1.
Authors’ description: From the 28 studies retrieved, miR-124 and FAM194/miR-124-2 methylation are the most frequently down- and up-regulated in CC progression, respectively. How The point of view quantitative demonstrated the quality of article research. Explanation necessary.
ANSWER 1. We thank the Reviewer for the suggestion. Quantitative amount of miRNA down/up regulation in terms of fold variation has been added in Table 2.
COMMENT 2. Add the PICO acronym (participants, interventions, comparisons, outcomes) in method section, for example, a table for eligibility criteria in accordance with PICO (participants, interventions, comparisons, outcomes).
ANSWER 2. We thank the Reviewer for the suggestion. A new Table reporting PICO elements for each one of the 28 examined studies has been added (Table 1).
COMMENT 3. Some words in the title “miRNAs and methylation” are not suitable as miRNA is a substantial but methylation is an event.
ANSWER 3. We thank the Reviewer for the note. The title has been corrected.

Reviewer 2 Report (New Reviewer)
Comments and Suggestions for Authors
The manuscript entitled “The Role of miRNAs and Methylation in HPV-Related Cervical Cancer: A Systematic Review” is a narrative review offering a comprehensive assessment of the epigenetic mechanisms involved in the molecular basis of cervical cancer. The topic is within the scope of the journal and would be interesting to clinicians and researchers engaged in cervical cancer management and research.
The manuscript is not well structured and requires major editing. There are numerous corrections that should be made and at various points clarifications are required (in order of appearance):
- The title is not well defined. Methylation of which molecules?
- In the abstract section, line 14, there is a statement about the microRNA-based mechanisms. Since this article also focuses on DNA methylation, it would be better to rephrase this sentence in order to include background information about DNA methylation as well. It is not well explained that methylation refers to microRNA-related loci. Stated I this manner, it seems that methylation of microRNA was assessed (like m6A methylation, etc).
- Line 22 – either list all the databases or remove “such as Pubmed”. Lines 22 and 28 are conflicting.
- The Introduction section and the most of the material and methods are focused on the functions and the expression of microRNA molecules. It is not clear how the results refer to DNA methylation and which methodology was used for assessing the contribution of this kind of epigenetic mechanisms on the molecular pathogenesis of cervical cancer. Presented in this manner, the parts of the manuscript referring to DNA methylation seem as added and not well connected to other sections.
- The purpose of the manuscript should be stated at the end of the Introduction, not in the first paragraph before the basic background information.
- The second paragraph od the Introduction seems merely inserted without any connection to the surrounding information. CIN2 and CIN3 are not defined here and the relevance for the topic is not explained, while the paragraph seems misplaced.
- line 175-181 seem irrelevant for microRNA functions.
-lines 219-236 belong to background, not to Results.
-section 2.2 is irrelevant for the topic of the manuscript. If microRNA mechanisms are in the focus, this section should not be in the Results. The parts of the Discussion corresponding to these results is also incorporated as an irrelevant addition.
- The discussion merely lists previous findings of experimental studies, without the evaluation, interpretation and critical appraisal. Significant improvement is necessary.
- The aims of the study are not well explained. If a methylation of DNA with a consequent microRNA expression dysregulation is the phenomenon intended to be evaluated, then the Title, Abstract and Introduction should be significantly rephrased and rewritten in order to highlight the major aims.
- What is the rationale for excluding the studies published prior to 2018? In that case, this is not a real systematic review with the stated topic. Moreover, a large number of studies were excluded based on this criterion.
- Data described in material and methods with the flow chart belong to the Results. Inclusion and exclusion criteria are not well elaborated. Was the study protocol registered in PROSPERO or other repository of predefined protocols for systematic reviews?
- Conclusions are merely a repetition of the background, not a summary of major findings from the review and the interpretation.
Author Response
COMMENT 1. The manuscript entitled “The Role of miRNAs and Methylation in HPV-Related Cervical Cancer: A Systematic Review” is a narrative review offering a comprehensive assessment of the epigenetic mechanisms involved in the molecular basis of cervical cancer. The topic is within the scope of the journal and would be interesting to clinicians and researchers engaged in cervical cancer management and research.
The manuscript is not well structured and requires major editing. There are numerous corrections that should be made and at various points clarifications are required (in order of appearance): The title is not well defined. Methylation of which molecules?
ANSWER 1. We thank the Reviewer for the note. The manuscript has been completely revised as suggested by the reviewer, in each step. The title has been changed.
COMMENT 2. In the abstract section, line 14, there is a statement about the microRNA-based mechanisms. Since this article also focuses on DNA methylation, it would be better to rephrase this sentence in order to include background information about DNA methylation as well. It is not well explained that methylation refers to microRNA-related loci. Stated I this manner, it seems that methylation of microRNA was assessed (like m6A methylation, etc).
ANSWER 2. In the abstract section, sentences have been rephrased adding background information dealing with DNA methylation (line 24,25 and 33,34).
COMMENT 3. Line 22 – either list all the databases or remove “such as Pubmed”. Lines 22 and 28 are conflicting.
ANSWER 3. The sentence has been removed as suggested.
COMMENT 4. The Introduction section and most of the material and methods are focused on the functions and the expression of microRNA molecules. It is not clear how the results refer to DNA methylation and which methodology was used for assessing the contribution of this kind of epigenetic mechanisms on the molecular pathogenesis of cervical cancer. Presented in this manner, the parts of the manuscript referring to DNA methylation seem as added and not well connected to other sections.
ANSWER 4. Information dealing DNA methylation has been added. As an examples in Table 2 ,19 out of the 28 reported papers deal with DNA methylation as now specified in the added column.
COMMENT 5. The purpose of the manuscript should be stated at the end of the Introduction, not in the first paragraph before the basic background information. The second paragraph od the Introduction seems merely inserted without any connection to the surrounding information. CIN2 and CIN3 are not defined here and the relevance for the topic is not explained, while the paragraph seems misplaced.
ANSWER 5. The introduction has been rewritten and rearranged as suggested.
COMMENT 6. line 175-181 seem irrelevant for microRNA functions.
ANSWER 6. The lines have been removed as suggested.
COMMENT 7. lines 219-236 belong to background, not to Results.
ANSWER 7. The lines have been moved to background as requested.
COMMENT 8. section 2.2 is irrelevant for the topic of the manuscript. If microRNA mechanisms are in the focus, this section should not be in the Results. The parts of the Discussion corresponding to these results is also incorporated as an irrelevant addition.
ANSWER 8. The oncoproteins has been added in the title, this aspect is always considered in the studies, and these are to be considered with epigenetic mechanisms.
COMMENT 9. The discussion merely lists previous findings of experimental studies, without the evaluation, interpretation, and critical appraisal. Significant improvement is necessary.
ANSWER 9. The discussion has been implemented with interpretation, and critical appraisal as suggested.
COMMENT 10. The aims of the study are not well explained. If a methylation of DNA with a consequent microRNA expression dysregulation is the phenomenon intended to be evaluated, then the Title, Abstract and Introduction should be significantly rephrased and rewritten to highlight the major aims.
ANSWER 10. We thank the Reviewer for the note .The Title, Abstract and Introduction have been rephrased and rewritten to highlight the major aims as suggested.
COMMENT 11. What is the rationale for excluding the studies published prior to 2018? In that case, this is not a real systematic review with the stated topic. Moreover, a large number of studies were excluded based on this criterion.
ANSWER 11. The lack of studies published before 2018, although justified by relevance maintenance, has been added as a limit of our study (Materials and Methods).
COMMENT 12. Data described in material and methods with the flow chart belong to the Results. Inclusion and exclusion criteria are not well elaborated. Was the study protocol registered in PROSPERO or other repository of predefined protocols for systematic reviews?
ANSWER 12. The flow chart has been moved to the Results as requested. The Inclusion and exclusion criteria have been better explained. The study protocol has been registered in Prospero.
COMMENT 13. Conclusions are merely a repetition of the background, not a summary of major findings from the review and the interpretation.
ANSWER 13. Conclusions have been changed and the major findings obtained has been added.
Reviewer 3 Report (New Reviewer)
Comments and Suggestions for Authors
The systematic review article "The Role of miRNAs and Methylation in HPV-Related Cervical Cancer: A Systematic Review," presented by Pulliero et al., focuses on the relationship between epigenetic changes and the progression of cervical cancer associated with human papillomavirus (HPV). It specifically examines the role of miRNA regulation and DNA methylation in the carcinogenesis of cervical cancer.
-The authors conducted a systematic literature search in major databases, such as PubMed, using predefined inclusion and exclusion criteria. Studies that investigated the expression, function, and clinical relevance of miRNAs in HPV-related cervical cancer were reviewed. It was identified that the dysregulation of miRNAs plays a crucial role in the pathogenesis of HPV-related cervical cancer. In particular, the methylation of FAM194/miR-124-2 emerged as a promising molecular marker for differentiating cases that require immediate surgical intervention from those that can be managed conservatively.
Major Comments
-The systematic review mentions that it followed the PRISMA guidelines for systematic reviews, but it would be beneficial to provide more details about the selection process, such as how many reviewers evaluated the studies and how disagreements were resolved. The search was limited to articles published between 2018 and 2023, which is justified to maintain relevance, but it could exclude important prior studies that might also add value to the topic.
-T1he review does not clearly mention whether a formal assessment of the quality of the included studies was conducted. Including a critical assessment of the methodological quality of the reviewed studies could strengthen the conclusions of the article. I recommend using validated tools, such as the Newcastle-Ottawa Scale or the QUADAS-2 tool, to evaluate the quality and risk of bias in studies of this type.
-I believe it would be helpful to expand the discussion on the heterogeneity among the reviewed studies. Were there significant differences in the techniques used to measure miRNA expression and DNA methylation? How are these differences interpreted in the context of current research?
-Although the clinical utility of FAM194/miR-124-2 methylation for guiding treatment is mentioned, the article could benefit from a more detailed discussion on how these observations could be integrated into daily clinical practice.
-The manuscript could improve by contextualizing its findings in relation to other previous studies and systematic reviews, discussing how its results provide new evidence or complement existing knowledge regarding the role of miRNAs and methylation in CC. A section exploring the gaps in current knowledge and future research directions necessary to advance the field is also lacking.
-I think the authors should delve deeper into the limitations of the included studies, such as the small sample size in some cases, which could affect the generalizability of the results. It would also be valuable to explore the potential limitations or challenges in implementing these biomarkers in the detection and management of CC in different clinical settings.
Minor Comments
-Acronyms should be defined; HPV is mentioned and then described again as human papillomavirus, the same applies to miRNAs - microRNA.
-Lines 44-63 in the introduction do not have references.
-Remove “we can divide” from line 95.
-It is evident that the introduction underwent modifications due to comments from other reviewers; however, it still needs to be restructured more appropriately.
-Lines 219-236 also lack references.
Comments on the Quality of English LanguageModerate editing of the English language is required.
Author Response
COMMENT 1. The systematic review mentions that it followed the PRISMA guidelines for systematic reviews, but it would be beneficial to provide more details about the selection process, such as how many reviewers evaluated the studies and how disagreements were resolved. The search was limited to articles published between 2018 and 2023, which is justified to maintain relevance, but it could exclude important prior studies that might also add value to the topic.
ANSWER 1. We thank the Reviewer for the note. Details dealing the selection process dealing the number of reviewers Two independent persons investigated the literature retrieved (G.C, P.A.) is now reported in Materials and Methods. A sentence dealing disagreement management has been added. The lack of studies published before 2018, although justified by relevance maintenance, has been added as a limit of our study (Materials and Methods).
COMMENT 2.
The review does not clearly mention whether a formal assessment of the quality of the included studies was conducted. Including a critical assessment of the methodological quality of the reviewed studies could strengthen the conclusions of the article. I recommend using validated tools, such as the Newcastle-Ottawa Scale or the QUADAS-2 tool, to evaluate the quality and risk of bias in studies of this type.
ANSWER 2.
We thank the Reviewer for the note. Dealing formal assessment of the quality of the included studies (a) PICO inclusion criteria for each study are now reported in Table 1; (b) a Table 1a reporting Newcastle-Ottawa Scale has been added (Table 1a) Methodological quality was evaluated using the Quality Assesment of Comparative Diagnostic Accuracy Studies (QUADAS-2) checklist.
COMMENT 3.
I believe it would be helpful to expand the discussion on the heterogeneity among the reviewed studies. Were there significant differences in the techniques used to measure miRNA expression and DNA methylation? How are these differences interpreted in the context of current research?
ANSWER 3. Thank you very much for your valuable advice. A paragraph discussing these points has been added in Discussion (lines 590-594).
COMMENT 4.
Although the clinical utility of FAM194/miR-124-2 methylation for guiding treatment is mentioned, the article could benefit from a more detailed discussion on how these observations could be integrated into daily clinical practice.
ANSWER 4. We thank the Reviewer for the note. In the discussion more details have been added in red color.
COMMENT 5.
The manuscript could improve by contextualizing its findings in relation to other previous studies and systematic reviews, discussing how its results provide new evidence or complement existing knowledge regarding the role of miRNAs and methylation in CC. A section exploring the gaps in current knowledge and future research directions necessary to advance the field is also lacking.
ANSWER 5. We thank the Reviewer for the note. The section with future research directions has been added in a new paragraph of the paper.
COMMENT 6.
I think the authors should delve deeper into the limitations of the included studies, such as the small sample size in some cases, which could affect the generalizability of the results. It would also be valuable to explore the potential limitations or challenges in implementing these biomarkers in the detection and management of CC in different clinical settings.
ANSWER 6. We thank the Reviewer for the comments. The limitations of the study have been included in the discussion section.
COMMENT 7.
Acronyms should be defined; HPV is mentioned and then described again as human papillomavirus, the same applies to miRNAs - microRNA.
ANSWER 7. We thank the Reviewer for the note. The Acronyms have been standardized.
COMMENT 8. Lines 44-63 in the introduction do not have references.
ANSWER 8. References in the Introduction have been added.
COMMENT 9. Remove “we can divide” from line 95.
ANSWER 9. The sentence it has been removed as suggested.
COMMENT 10. It is evident that the introduction underwent modifications due to comments from other reviewers; however, it still needs to be restructured more appropriately.
ANSWER 10. We thank the Reviewer for the note. The introduction has been revised
COMMENT 11. Lines 219-236 also lack references.
ANSWER 11. The references have been added.
COMMENT 12. Moderate editing of the English language is required.
ANSWER 12. We have requested the revision of English language by the MPDI Service.
Reviewer 4 Report (New Reviewer)
Comments and Suggestions for Authors
In this article, the authors presented the relationship between epigenetic changes in the development and progression of HPV associated with CC. The manuscript is straightforward, well written, and concise and has clear results within the scope of a systematic review. Definitely deserves to be published and is a valuable contribution to the “International journal of Molecular Sciences”. However, the following comments need to be addressed, as recommended.
[1] “1. Introduction”, Page 3 of 22, Lines 117-120:
“Over the past years, the molecular understanding of malignant transformation and epidemiologic information related to HPV has led to the development of many strategies based on new biomarkers aimed at a better detection and early intervention in patients with cervical dysplasia and CC [11,14].”.
In addition to squamous cell carcinoma, CC can also present as adenocarcinoma, which accounts for approximately 20–25% of all cervical malignancies. From a molecular and prognostic perspective, authors should mention that PD-L1 expressions are more frequent in squamous cell carcinoma than in adenocarcinoma. In squamous cell carcinoma, diffuse PD-L1 expressions are associated with poorer disease-free survival and disease-specific survival, while marginal PD-L1 expressions are linked to a significantly more favorable prognosis. In adenocarcinoma, patients whose tumors lack PD-L1-positive tumor-associated macrophages demonstrate a survival benefit.
Recommended reference: Chakravarthy A, et al. Integrated analysis of cervical squamous cell carcinoma cohorts from three continents reveals conserved subtypes of prognostic significance. Nat Commun. 2022 Oct 7;13(1):5818.
[2] “1. Introduction”, Page 4 of 22, Lines 164-166:
“Therefore, miRNAs are potential candidates in oncology as diagnostic biomarkers, prognostic biomarkers, therapeutic targets, and preventive screening programs [20].”.
At that point, the authors should mention that exosomes have emerged as a novel source of non-invasive tumor biomarkers. The unique bilayer membrane structure of exosomes offers protection against external RNases and proteases, leading to enhanced stability of the enclosed mRNAs, miRNAs, and functional proteins, thus making exosomes highly sensitive markers for disease diagnosis. The cargo in tumor-derived exosomes, such as the range of miRNAs, can also serve as biomarkers, offering valuable targets for early detection, diagnosis and treatment.
Recommended reference: Boussios S, et al. Exosomes in the Diagnosis and Treatment of Renal Cell Cancer. Int J Mol Sci. 2023;24(18):14356.
Author Response
COMMENT 1. Introduction”, Page 3 of 22, Lines 117-120:
“Over the past years, the molecular understanding of malignant transformation and epidemiologic information related to HPV has led to the development of many strategies based on new biomarkers aimed at a better detection and early intervention in patients with cervical dysplasia and CC [11,14].”.
In addition to squamous cell carcinoma, CC can also present as adenocarcinoma, which accounts for approximately 20–25% of all cervical malignancies. From a molecular and prognostic perspective, authors should mention that PD-L1 expressions are more frequent in squamous cell carcinoma than in adenocarcinoma. In squamous cell carcinoma, diffuse PD-L1 expressions are associated with poorer disease-free survival and disease-specific survival, while marginal PD-L1 expressions are linked to a significantly more favorable prognosis. In adenocarcinoma, patients whose tumors lack PD-L1-positive tumor-associated macrophages demonstrate a survival benefit.
Recommended reference: Chakravarthy A, et al. Integrated analysis of cervical squamous cell carcinoma cohorts from three continents reveals conserved subtypes of prognostic significance. Nat Commun. 2022 Oct 7;13(1):5818.
ANSWER 1. We thank the Reviewer for the note. The reference has been added.
COMMENT 2. Introduction”, Page 4 of 22, Lines 164-166:
“Therefore, miRNAs are potential candidates in oncology as diagnostic biomarkers, prognostic biomarkers, therapeutic targets, and preventive screening programs [20].”.
At that point, the authors should mention that exosomes have emerged as a novel source of non-invasive tumor biomarkers. The unique bilayer membrane structure of exosomes offers protection against external RNases and proteases, leading to enhanced stability of the enclosed mRNAs, miRNAs, and functional proteins, thus making exosomes highly sensitive markers for disease diagnosis. The cargo in tumor-derived exosomes, such as the range of miRNAs, can also serve as biomarkers, offering valuable targets for early detection, diagnosis and treatment.
Recommended reference: Boussios S, et al. Exosomes in the Diagnosis and Treatment of Renal Cell Cancer. Int J Mol Sci. 2023;24(18):14356.
ANSWER 2. We thank the Reviewer for the note. The reference has been added.

Round 2
Reviewer 2 Report (New Reviewer)
Comments and Suggestions for Authors
The authors have made some corrections that improved the quality of the manuscript. However, the this format of the manuscript is more appropriate for a narrative review, not for systematic review of any kind, since there are some parts of the manuscript that are not focused on the narrow topic and are more appropriate for the background, while the Material and Methods should be elaborated in detail.
The corrections in Material and Methods are not sufficient and additional information is needed. PROSPERO registration number should be stated. I further disagree with the response to Comment 8. I am not sure how the authors made corrections regarding the Comment 5, since I do not see any corrections corresponding to this comment.
Author Response
Reviewer 2:
Comment 1:
The authors have made some corrections that improved the quality of the manuscript. However, the this format of the manuscript is more appropriate for a narrative review, not for systematic review of any kind, since there are some parts of the manuscript that are not focused on the narrow topic and are more appropriate for the background, while the Material and Methods should be elaborated in detail.
The corrections in Material and Methods are not sufficient and additional information is needed. PROSPERO registration number should be stated. I further disagree with the response to Comment 8. I am not sure how the authors made corrections regarding the Comment 5, since I do not see any corrections corresponding to this comment.
ANSWER 1. We thank the Reviewer for the suggestion. The manuscript has been improved as suggested. The Material and Methods have been implemented. The PROSPERO registration number will be given in a few days, we will indicate it in the paper as soon as it is available.
COMMENT 5.
The purpose of the manuscript should be stated at the end of the Introduction, not in the first paragraph before the basic background information. The second paragraph of the Introduction seems merely inserted without any connection to the surrounding information. CIN2 and CIN3 are not defined here and the relevance for the topic is not explained, while the paragraph seems misplaced.
ANSWER 5. We thank the Reviewer for the note. The purpose of the manuscript has been added at the end of the Introduction. The second paragraph has been re -written. CIN2 and CIN3 have been defined as suggested.
COMMENT 8. section 2.2 is irrelevant for the topic of the manuscript. If microRNA mechanisms are in the focus, this section should not be in the Results. The parts of the Discussion corresponding to these results is also incorporated as an irrelevant addition.
ANSWER 8. We thank the Reviewer for the suggestion. The section 2.2 has been removed and it has been considered the microRNA and methylation mechanisms as suggested by the reviewer. Addictionally, the parts of the Discussion corresponding to these results (relating section 2.2) have been removed. The references have been listed again.

Reviewer 3 Report (New Reviewer)
Comments and Suggestions for Authors
I consider that the authors have made substantial changes to their manuscript, the revision can be published.
Please review the legend in Figure 2 to deliver a clear message and check spelling like in Figure 2 and 3. More detailed descriptions are needed. The figure legends must be further refined to clearly explain each figure, provide a detailed but simple guided description, and highlight the significance of the findings.
Author Response
Reviewer 3:
COMMENT 1:
I consider that the authors have made substantial changes to their manuscript, the revision can be published.
Please review the legend in Figure 2 to deliver a clear message and check spelling like in Figure 2 and 3. More detailed descriptions are needed. The figure legends must be further refined to clearly explain each figure, provide a detailed but simple guided description, and highlight the significance of the findings.
ANSWER 1. We thank the Reviewer for the suggestion. The figure legend of the Figure 2 and 3 have been implemented with more detailed as suggested.

This manuscript is a resubmission of an earlier submission. The following is a list of the peer review reports and author responses from that submission.
Round 1
Reviewer 1 Report
Comments and Suggestions for Authors
This review provides an up-to-date literature regarding the role of micro RNA and DNA methylation in HPV related CC. It is an interesting manuscript. The introduction section should be shortened. In the introduction section the progression and regression rates for CIN2 and CIN3 should be provided separately (lines 76-77). I would suggest the authors to shorten the discussion section and not describe all different studies separately. At the end I would suggest adding two short paragraphs to provide a summary with regards to what the findings were from the host DNA and from HPV DNA studies. Finally, when mentioning the role of methylation in conservative management of CIN2 and CIN3 (Kremer et al, lines 363-368) the study by Louvanto et al. (PMID 31344232) should be mentioned
Author Response
Dear Editors,
We would like to thank you for considering the manuscript entitled “The Role of miRNAs and Methylation in HPV-Related Cervical Cancer: A Systematic Review” by Cassatella G. et al., and for sharing the Reviewers’ comments that certainly helped in improving the quality of the manuscript (ID: ijms-3036094). We appreciated the Reviewers’ comments, and we revised the manuscript accordingly. Please find enclosed to the submission of the revised version of the manuscript the point-by point reply to the Reviewers’ comments. For clarity’s sake, changes in the revised MS are wrote in yellow color.
We hope that this revised version of our MS will be now suitable for publication in the IJMS.
Accordingly, we prepared a revised version of the manuscript acknowledging Referees’ and Editor’s comments as below specified:
Reviewer 1
COMMENT 1.
This review provides an up-to-date literature regarding the role of microRNA and DNA methylation in HPV related CC. It is an interesting manuscript. The introduction section should be shortened. In the introduction section the progression and regression rates for CIN2 and CIN3 should be provided separately (lines 76-77). I would suggest the authors to shorten the discussion section and not describe all different studies separately. At the end I would suggest adding two short paragraphs to provide a summary with regards to what the findings were from the host DNA and from HPV DNA studies. Finally, when mentioning the role of methylation in conservative management of CIN2 and CIN3 (Kremer et al, lines 363-368) the study by Louvanto et al. (PMID 31344232) should be mentioned.
ANSWER 1. We thank the Reviewer for the suggestion. The introduction has been shortened accordingly to the reviewer suggestions. Additionally, the discussion section is reorganized as suggested by the reviewers. Two short paragraphs have been added, and the reference mentioned has been added.
Reviewer 2 Report
Comments and Suggestions for Authors
In the manuscript “The Role of miRNAs and Methylation in HPV-Related Cervical Cancer: A Systematic Review,” the authors have comprehensively analyzed the existing literature to clarify the intricate relationships between epigenetic markers modulation and the development and progression of HPV-associated cervical cancer.
The data presented herein are valuable and engaging.
The article is written correctly, adequately illustrated, and supplemented with informative tabular data. The findings collected in the study were sufficiently explained and discussed. A corresponding and up-to-date reference list accompanies the manuscript.
I have no special remarks or requests.
The acceptance of the manuscript in its current form is suggested.
Author Response
Reviewer 2
COMMENT 1.
In the manuscript “The Role of miRNAs and Methylation in HPV-Related Cervical Cancer: A Systematic Review,” the authors have comprehensively analyzed the existing literature to clarify the intricate relationships between epigenetic markers modulation and the development and progression of HPV-associated cervical cancer.
The data presented herein are valuable and engaging.
The article is written correctly, adequately illustrated, and supplemented with informative tabular data. The findings collected in the study were sufficiently explained and discussed. A corresponding and up-to-date reference list accompanies the manuscript.
I have no special remarks or requests.
The acceptance of the manuscript in its current form is suggested.
ANSWER 1. We thank the Reviewer for the positive comments.
Reviewer 3 Report
Comments and Suggestions for Authors
The authors conduct a systematic review of 28 articles on the role of methylation and regulation in miRNA expression in cervical cancer associated with HPV infection.
1-As indicated in the abstract, the review aims to examine the relationship between epigenetic changes in the development and progression of HPV associated with cervical cancer. Despite its indications, neither aspect is addressed throughout. The document.
2-I recommend that the authors focus on the introduction, eliminating redundancies regarding the description of the virus and its constituents.
3- Several concepts of molecular biology and biochemistry lack foundation, so I recommend that the authors contact an expert in epigenetics who can clarify poorly raised concepts about the nature of miRNAs and their classifications.
4- Line 119. review was to identify. Line 21, Cervical Cancer
5- line 74-86, bibliographic references are missing
6- Table 1. The classification generates confusion. The authors indicate one column for proteins and another for genes. So, where are the miRNAs located? And the methylations? It should be noted that there are methylation and demethylation, so in the last column, where they indicate up and downregulation, what do they refer to? To an increase in methylation and a decrease in methylation. An increase in methylation of the gene that encodes a particular protein indicates silencing of that gene (downregulation). On the contrary, an upregulation of miRNA indicates increased gene expression. The authors' Table 1 that they have developed generates a lot of confusion and misinterpretation of the facts. I recommend that both aspects be treated separately.
7- The nomenclature for genes and proteins is not correctly indicated in the manuscript, which leads to continuous confusion.
8. Fig 1. Only 25 articles are indicated. Perhaps it is more convenient that the clinical aspects of the articles analyzed are all indicated in the same table.
9. Fig 2. What value does knowing the types of studies (histologous, cytological...) give to the review if what would really matter would be the molecular technique used to detect miRNAs and methylations.
10. Fig. 4. The number of articles and the oncoproteins analyzed have no scientific value. In this case there are 4 articles analyzed.
11. Fig. 5 Idem. They cannot base the results on the frequency of appearance of articles.
Author Response
Reviewer 3
COMMENT 1.
The authors conduct a systematic review of 28 articles on the role of methylation and regulation in miRNA expression in cervical cancer associated with HPV infection.
As indicated in the abstract, the review aims to examine the relationship between epigenetic changes in the development and progression of HPV associated with cervical cancer. Despite its indications, neither aspect is addressed throughout. The document.
ANSWER 1. We thank the Reviewer for the note. We sincerely thank the reviewer for careful reading. After following your comments, we have revised the manuscript point out the relationship between epigenetic changes in the development and progression of HPV associated with cervical cancer.
COMMENT 2.
I recommend that the authors focus on the introduction, eliminating redundancies regarding the description of the virus and its constituents.
ANSWER 2. We thank the Reviewer for the note. We have been revised the introduction according to the suggestions.
COMMENT 3.
Several concepts of molecular biology and biochemistry lack foundation, so I recommend that the authors contact an expert in epigenetics who can clarify poorly raised concepts about the nature of miRNAs and their classifications.
ANSWER 3. Thank you very much for your valuable advice. We made the corresponding changes in the revised manuscript.
COMMENT 4.
Line 119. review was to identify. Line 21, Cervical Cancer
ANSWER 4. We thank the Reviewer for the note. We have been revised the manuscript as suggested.
COMMENT 5.
line 74-86, bibliographic references are missing
ANSWER 5. Thank you very much for your precious advice. The references have been added.
COMMENT 6.
Table 1. The classification generates confusion. The authors indicate one column for proteins and another for genes. So, where are the miRNAs located? And the methylations? It should be noted that there are methylation and demethylation, so in the last column, where they indicate up and downregulation, what do they refer to? To an increase in methylation and a decrease in methylation. An increase in methylation of the gene that encodes a particular protein indicates silencing of that gene (downregulation). On the contrary, an upregulation of miRNA indicates increased gene expression. The authors' Table 1 that they have developed generates a lot of confusion and misinterpretation of the facts. I recommend that both aspects be treated separately.
ANSWER 6. We thank the Reviewer for the note. The Table 1 it has been reformulated as suggested.
COMMENT 7.
The nomenclature for genes and proteins is not correctly indicated in the manuscript, which leads to continuous confusion.
ANSWER 7. We thank the Reviewer for the note. We have corrected the nomenclature.
COMMENT 8.
Fig 1. Only 25 articles are indicated. Perhaps it is more convenient that the clinical aspects of the articles analyzed are all indicated in the same table.
ANSWER 8. We thank the reviewer for the annotation. The Table 1 it has been revised as suggested adding the clinical aspects.
COMMENT 9.
Fig 2. What value does knowing the types of studies (histologous, cytological...) give to the review if what would really matter would be the molecular technique used to detect miRNAs and methylations.
ANSWER 9. Thank you very much for your valuable advice. Fig 2 has been removed because the histologist aspect is not the focus of the paper, but the epigenetic control is the important aspect of this review paper.
COMMENT 10.
Fig. 4. The number of articles and the oncoproteins analyzed have no scientific value. In this case there are 4 articles analyzed.
ANSWER 10. Thank you very much for your valuable advice. The Figure 4 has been deleted. We have created a new table with the information of the oncoproteins and HPV reported in the studies.
COMMENT 11.
Fig. 5 Idem. They cannot base the results on the frequency of appearance of articles.
ANSWER 11. Thank you very much for your valuable advice. The Figure 5 has been deleted. We have created a new table with the information of the microRNAs and HPV reported in the studies.
Round 2
Reviewer 3 Report
Comments and Suggestions for Authors
The authors have partially responded to the questions raised. This manuscript still needs a thorough revision in many aspects.
Some issues remain unresolved:
1. The manuscript is blurred when dealing with two aspects of regulation, such as miRNA and methylation. Throughout the text, both concepts are intertwined.
2. The objective indicated in the abstract indicates that the relationship between epigenetic modulation and, on the one hand, development and progression of CHD will be studied. These last two aspects are not well developed in the text and should be separated and studied separately.
3. line 16, CC, cervical cancer, is cited in the text for the first time.
4. The results of the abstract only indicate data observed in CC progression.
5. The introduction has improved but could be more focused.
6. Line 106. Many Studies. The authors only indicate a reference.
7. At the end of the introduction, the authors try to specify the study's objective without achieving it. Again, there is redundancy in aspects already discussed in the introduction.
8. All tables and figures should be explained, and different sections should be based on the results of the tables.
9. Table 1 is a mess; the authors should rethink the organization of this table. It does NOT make sense to talk about methylation in the column of proposed biomarkers "methylation" and "classifier"; epigenetic regulation already has a column called regulation. The number of subjects should be in the second column. It is NOT considered when it is CHD development and when it is CHD progression in Table 1.
10. Table 1 does not help to understand the results. No abbreviations are included in the table footer.
11. Fig 1 and Fig 2. Knowing the number of articles in both variables has no scientific meaning. If the authors want to make any mention in the text, highlighting these aspects in the manuscript does not make sense.
12. Table 2. The function of the oncoproteins analyzed is not shown as indicated in the title of Table 2. The methodology used for their analysis and the sample type are observed in some cases. The table is confusing and underdeveloped.
13. The authors should explain the most important results in Table 3.
14. The study is not registered in PROSPERO
15. In the methodology, when the 28 articles are selected, a table should be placed below showing those 28 articles, sample sizes, biomarkers, etc., so the authors would not have to write redundant data as shown in the manuscript
The work needs to be focused and reorganized to have clinical value.